# Microarray Technology May Reveal the Contribution of Allergen Exposure and Rhinovirus Infections as Possible Triggers for Acute Wheezing Attacks in Preschool Children

**DOI:** 10.3390/v13050915

**Published:** 2021-05-15

**Authors:** Katarzyna Niespodziana, Katarina Stenberg-Hammar, Nikolaos G. Papadopoulos, Margarete Focke-Tejkl, Peter Errhalt, Jon R. Konradsen, Cilla Söderhäll, Marianne van Hage, Gunilla Hedlin, Rudolf Valenta

**Affiliations:** 1Division of Immunopathology, Department of Pathophysiology and Allergy Research, Center of Pathophysiology, Infectiology and Immunology, Medical University of Vienna, 1090 Vienna, Austria; katarzyna.niespodziana@meduniwien.ac.at (K.N.); margarete.focke-tejkl@meduniwien.ac.at (M.F.-T.); 2Astrid Lindgren Children’s Hospital, Karolinska University Hospital, 171 77 Stockholm, Sweden; Katarina.stenberg@bsmartina.se (K.S.-H.); Jon.konradsen@ki.se (J.R.K.); cilla.soderhall@ki.se (C.S.); gunilla.hedlin@ki.se (G.H.); 3Department of Women’s and Children’s Health, Karolinska Institutet, 171 77 Stockholm, Sweden; 4Division of Infection, Immunity & Respiratory Medicine, University of Manchester, Manchester M13 9NT, UK; ngpallergy@gmail.com; 5Allergy Department, 2nd Pediatric Clinic, University of Athens, 106 79 Athens, Greece; 6Karl Landsteiner University for Healthcare Sciences, 3500 Krems, Austria; 7Department of Pneumology, University Hospital Krems, Austria, and Karl Landsteiner University of Health Sciences, 3500 Krems, Austria; Peter.Errhalt@krems.lknoe.at; 8Division of Immunology and Allergy, Department of Medicine, Solna, Karolinska Institutet and Karolinska University Hospital, 171 77 Stockholm, Sweden; marianne.van.hage@ki.se; 9National Research Center, Institute of immunology, FMBA of Russia, 115478 Moscow, Russia; 10Laboratory of Immunopathology, Department of Clinical Immunology and Allergy, Sechenov First Moscow State Medical University, 119435 Moscow, Russia

**Keywords:** rhinovirus, wheeze, asthma, preschool children, allergy, allergen, IgE sensitization, allergen exposure, microarray

## Abstract

Allergen exposure and rhinovirus (RV) infections are common triggers of acute wheezing exacerbations in early childhood. The identification of such trigger factors is difficult but may have therapeutic implications. Increases of IgE and IgG in sera, were shown against allergens and the N-terminal portion of the VP1 proteins of RV species, respectively, several weeks after allergen exposure or RV infection. Hence, increases in VP1-specific IgG and in allergen-specific IgE may serve as biomarkers for RV infections or allergen exposure. The MeDALL-allergen chip containing comprehensive panels of allergens and the PreDicta RV chip equipped with VP1-derived peptides, representative of three genetic RV species, were used to measure allergen-specific IgE levels and RV-species-specific IgG levels in sera obtained from 120 preschool children at the time of an acute wheezing attack and convalescence. Nearly 20% of the children (22/120) showed specific IgE sensitizations to at least one of the allergen molecules on the MeDALL chip. For 87% of the children, increases in RV-specific IgG could be detected in the follow-up sera. This percentage of RV-specific IgG increases was equal in IgE-positive and -negative children. In 10% of the children, increases or *de novo* appearances of IgE sensitizations indicative of allergen exposure could be detected. Our results suggest that, in the majority of preschool children, RV infections trigger wheezing attacks, but, in addition, allergen exposure seems to play a role as a trigger factor. RV-induced wheezing attacks occur in IgE-sensitized and non-IgE-sensitized children, indicating that allergic sensitization is not a prerequisite for RV-induced wheeze.

## 1. Introduction

Allergen exposure in sensitized individuals and rhinovirus (RV) infections are common triggers for acute wheezing attacks during early childhood [1] and for exacerbations of respiratory diseases in adults [2]. For children aged three years or older, RV accounts for 75% to 80% of virus-induced attacks that lead to hospitalizations, and the majority of these children have been reported to be atopic [3]. Several studies have indicated that RV-induced asthma and wheezing are more common and/or more severe in allergic subjects [1,4], and that this could be due to Th2 features of the allergic immune response [5]. It was also reported that allergic sensitization is particularly associated with RV-induced wheezing in children [6]. Furthermore, levels of allergic sensitization correlate well with clinical outcomes [7]. Conversely, viral infections were found to be the dominant risk factors for wheezing in children before three years of age, irrespective of atopy [1,8,9]. In this context, birth cohort studies have shown that allergic sensitization is less common in the first few years of life, whereas it becomes more frequent at school age and later in adolescence [10].

RV infections and exposure to inhaled allergens in sensitized patients appear to increase the risk of developing asthma, suggesting that both factors may act synergistically to raise the risk of wheeze/asthma exacerbations [2,6,11]. In fact, wheezing episodes that are caused by RVs in children less than three years of age augment the risk of the subsequent development of asthma [12]. In this context, several possibilities may be considered in regard to how allergic sensitization and RV infections can act synergistically to promote the development of asthma [13]. One possibility is that a Th2-prone immune system may be less effective in defending against RV infections [5]. On the contrary, it was shown that rhinovirus infections may induce damage to the respiratory epithelial barrier and, thus, may facilitate the penetration of allergens into the sub-epithelial tissues, where they may cause an increased allergic inflammation [14,15,16].

It is, therefore, important to have diagnostics that allow the complete mapping of the molecular sensitization profiles of allergic patients, as well as the measurement of the effects of allergen exposure that cause symptoms via serology. For this purpose, chips containing comprehensive panels of micro-arrayed allergen molecules were developed [17,18,19]. Furthermore, it was shown that the allergen exposure that led to clinical reactions induced detectable increases in allergen-specific IgE in the serum of the exposed patient a few weeks after the allergen exposure had occurred [20].

RV infections capable of triggering respiratory symptoms in infected subjects can be traced in a species-specific manner using synthetic peptides derived from the N-terminus of the VP1 capsid protein of different RV strains either by ELISA or by using micro-arrayed VP-1-derived peptides from the most common RV strains [21,22,23,24]. In fact, it was demonstrated that RV-triggered asthma exacerbations lead to increases in IgG levels specific to the VP1-derived peptides of the culprit RV strains [22,23,24,25]. This can be measured by comparing specific IgG levels in a serum sample obtained at the time of RV infection, triggering the exacerbation of asthma with the levels in a sample obtained approximately ten weeks later. Accordingly, the use of these biomarkers was proposed for the differential diagnosis of respiratory disease exacerbations triggered by allergen exposure in a sensitized subject and/or by RV infections to initiate personalized forms of asthma treatment based on the identification of the asthma trigger factors [26]. In this study, we applied this differential diagnostic approach in a cohort of 120 preschool children who had been admitted to the hospital due to an acute wheezing attack [23,27]. Serum samples were taken during the acute visit and several weeks after, and IgE reactivity profiles to multiple allergens, as well as IgG responses to a panel of N terminal VP1 peptides of different RV strains, were measured to identify the cause of these wheezing attacks. Thus, the hypothesis of this study was that increases in allergen-specific serum IgE levels and/or increases in RV-specific IgG levels in follow-up serum samples are indicative of allergen exposure and/or RV infections as possible trigger factors for the wheezing attack. Furthermore, the comprehensive analysis of IgE sensitizations would inform whether IgE-sensitized children are more likely to experience RV-triggered wheezing attacks.

Therefore, this study is the first to relate increases in allergen-specific IgE levels and rhinovirus-specific antibodies measured with defined allergen molecules and defined peptides from RV coat proteins as indicators for specific immune responses in children with documented wheezing episodes. Accordingly, serum samples were obtained at the time point when the wheezing episode occurred in these children and some weeks later in order to be able to assess whether allergen contact and/or an RV infection had boosted the specific antibody responses. Furthermore, the comprehensive analysis of IgE sensitizations in the children with microarrayed allergens allowed the determination of the frequency of RV-triggered wheezing attacks in IgE-sensitized versus non-sensitized children.

## 2. Methods

### 2.1. Subjects and Sample Collection

We analyzed serum samples of 120 children, aged 6 months to 42 months, who had been admitted to the Children’s Emergency Wards, Astrid Lindgren Children’s Hospital, Sweden, due to acute symptoms of wheeze [23,24,27]. Peripheral blood samples were acquired during the emergency and follow-up visits between 6 and 30 weeks (median of 11 weeks) after the initial recruitment. Nasopharyngeal swab samples were also obtained from these children during the acute and follow-up visits. In 118 of the 120 children, a molecular diagnostic platform for the rapid detection of 15 respiratory strains was used, and the following respiratory viruses were detected: Adenovirus: 7 children; Bocavirus: 8 children; Coronavirus: 6 children; Influenza A/B: 1 child; Metapneumovirus: 3 children; Parainfluenza virus: 4 children; RSV: 22 children [23]. In 108 of the 120 children, RV strains were identified by nested PCR targeting VP4/VP2 region and by subsequent sequencing, as previously described [24]. Written informed consents were obtained from their legal guardians [23,24,27], and the analysis of pseudonymized samples was performed at the Medical University of Vienna with permission from the Ethics committee of the Medical University of Vienna EK (EK1721/2014).

To demonstrate that increases in allergen-specific IgE levels can be detected by micro-array analysis, sera from 21 representative grass pollen allergic patients were randomly picked from a placebo-treated group of a double-blind, placebo-controlled, multicenter clinical trial conducted from May 2012 to October 2014 (ClinicalTrial.gov Identifier NCT01538979) [28]. Pseudonymized samples were analyzed for allergen-specific IgE reactivity with permission from the Ethics Committee of Medical University of Vienna (EK 1641/2014).

### 2.2. Determination of Antibody Reactivity Profiles by MeDALL Allergen- and PreDicta RV-Chip

Allergen-specific IgE levels in serum samples from 120 children collected during the acute and follow-up visits were determined by the MeDALL-allergen chip, which comprises more than 170 micro-arrayed allergens and utilizes only minute amounts of serum (Phadia-Thermo Fisher, Uppsala, Sweden) [18]. Allergen-specific IgE levels were measured by triplicate determinations and are expressed as median values, as recommended by the manufacturer’s instructions for ISAC 112 (Thermo Fisher Scientific, Uppsala, Sweden). Microarrays were washed exclusively with a washing solution and were dried by centrifugation (1 min; 1000× *g*; RT) [18]. RV-specific IgG responses were measured using a PreDicta RV-chip containing 30 N terminal VP1 peptides from different RV strains representing the three known genetic species: RV-A, RV-B, and RV-C [24]. Figure 1 shows the highest RV strain-specific antibody increases measured during the acute and follow-up visits for each child. For children presenting with mixed RV infections, the highest RV strain (RV-A or RV-C)-specific increases are shown. RV-specific IgGs were measured and analyzed as described previously [24]. Levels of allergen-specific IgEs and RV-specific IgGs were reported in ISAC-standardized units (i.e., ISU-E and ISU-G) with cut-off values of 0.3 and 1 ISU, respectively. For the calibration and detection of background signals, a calibrator serum and a sample diluent were included in each analysis of the run [18,24].

### 2.3. Statistical Analysis

Differences in antibody responses were calculated using a signed-rank Wilcoxon test for evaluating paired differences between two time points (i.e., the acute and follow-up visits and before and after grass pollen season) within each analysis group using GraphPad Prism 6 software (La Jolla, CA, USA). *p* values < 0.05 were considered significant.

## 3. Results

### 3.1. The Majority of Pre-School Children Presenting with Acute Wheezing Attacks Show RV Species-Specific IgG Increases

The measurement of RV-species-specific IgG increases after controlled RV-infection or RV-triggered wheezing attacks has already been demonstrated in several studies [22,23,24]. Figure 1 shows the results of the analysis of sera from 120 pre-school children presenting with acute wheezing attacks (Table 1) regarding increases in IgG antibody levels specific to RV strain-specific peptides in follow-up blood samples, as tested by the PreDicta RV-chip. This chip contained 30 N-terminal VP1 peptides, including 18 peptides of RV-A species, 9 peptides of RV-B, and 3 peptides of RV-C species. As peptides from RV strains belonging to the same RV species show a higher sequence similarity than those from unrelated species, the cross-reactivity to peptides from one species can be observed, which allows the establishment of cumulative species-specific antibody levels [25]. Thus, the RV strain with the highest increase in VP1-specific IgG in the follow-up blood sample was assumed to indicate the culprit RV species (Figure 1A,B) [24]. Significant increases in RV peptide-specific IgG antibodies were observed during the follow-up visit in 104 out of the 120 tested children (87%), indicating that RV infections may have triggered the wheezing attack in the majority of the children (Figure 1A).

The increases in IgG levels to species-specific peptides were as follows: RV-A-acute visit median ISU-G: 14.6, follow-up visit median ISU-G: 44.4; RV-B-acute visit median ISU-G: 5.0, follow-up visit median ISU-G: 35.5; RV-C-acute visit median ISU-G: 18.9, follow-up visit median ISU-G: 67.8; RV-mixed-acute visit median ISU-G: 16.7, follow-up visit median ISU-G: 37.5 and, thus, significant (Figure 1B). IgG increases occurred most frequently to peptides from RV-A (34%) and RV-C (27%) species, followed by RV-B species (19%) (Figure 1A,B). IgG increases to peptides from at least two different RV species were found in 6% of the children (Figure 1B), indicating a mixed or subsequent infection with different RV strains. For sixteen children (i.e., 13%), we did not observe relevant increases in RV-specific IgG responses. Next, we analyzed increases of RV-species-specific antibodies in children who had their follow-up visits <10 weeks after the wheezing attack (group 1), 10–14 weeks after the wheezing attack (group 2) and >14 weeks after the wheezing attack (group 3) (Figure 2). Although not reaching statistical significance, this analysis showed that the rises of RV-group-specific antibodies in convalescent sera were highest in group 3, followed by group 2 and group 1. Accordingly, 10 weeks was a suitable time for measuring the antibody responses to RV infections (Figure 2).

### 3.2. Role of Allergen-Specific IgE sensitizations and allergen Exposure as Possible Trigger Factors for Acute Wheezing in Certain Pre-School Children

We performed an analysis of allergic sensitizations to a comprehensive panel of allergen molecules comprising the most common allergen sources in our cohort using the MeDALL allergen chip. The MeDALL allergen chip contains more than 160 micro-arrayed allergen molecules comprising the most common allergen sources (Appendix A) [18]. We found that 22 out of the 120 children (i.e., 18%) showed specific IgE sensitizations to at least one of the allergen molecules on the MeDALL chip (Figure 3 and Figure 4A). The percentage of IgE-sensitized children was lowest in the wheezing children aged 6–12 months (i.e., 14.3%) and increased with the children higher in age (age 13–24 months: 15.1%; age 25–42 months: 29%) (Table 1, Figure 4B).

Twenty-one out of the 22 IgE-sensitized children exhibited IgE reactivities to food allergens (i.e., 95%), particularly to allergen molecules from peanuts (*n* = 8), cow’s milk (*n* = 7), and eggs (*n* = 12) (Figure 3). Ten out of the 22 children (i.e., 41%) were sensitized to respiratory allergens, including allergens from birch (Bet v 1: *n* = 3), dogs (Can f 1: *n* = 3), and cats (Fel d 1: *n* = 3) (Figure 3).

### 3.3. RV-Induced Wheeze Attacks Occur at the Same Frequencies in IgE-Sensitized, as Well as in Nonsensitized, Pre-School Children

The population of preschool children experiencing an acute wheeze attack consisted of children of different ages (Table 1). One group consisted of children aged 6–12 months (*n* = 35); in the second group, the age was 13–24 months (*n* = 53); in the third group, the age was 25–42 months (*n* = 31) (Table 1). The children in our study were, thus, younger than 4 years and showed relatively low frequencies of IgE sensitizations (Figure 4B). The IgE sensitization rate in the youngest group of children was 14.3%, increasing by age to 15.1% in children aged 13–24 months, and reaching 29% in children aged 25–42 months (Figure 4B and Table 1).

Interestingly, when we compared the percentages of children with RV-specific antibody increases indicative of RV-triggered wheeze in the children with and without IgE sensitizations, we found an equal percentage of children with RV-induced wheeze (i.e., 86% and 87%, respectively) (Figure 4A). Thus, there was no evidence that RV-induced wheeze is more common in IgE-sensitized children as compared to non-IgE-sensitized children.

### 3.4. Evidence for Allergen Exposure as Trigger Factor for Early Childhood Wheezing

In order to investigate whether the acute wheezing attack might have been triggered by allergen contact, we also measured IgE levels in follow-up sera collected several weeks later (median of 11 weeks; Table 1), when no sign of infection was present, as examined by a study physician and by PCR-based testing [23,29]. The collection of evidence for the absence of infection during the follow-up visit was relevant to reduce the possibility of the occurrence of infections with additional viruses in the convalescence period.

In fact, several studies have demonstrated that allergen contact via the respiratory mucosa strongly boosts allergen-specific IgE levels in the blood [20,30,31,32]. For example, Niederberger et al. [20] showed that specific serum IgE continue to rise until five weeks after a single nasal exposure to the allergen molecule, which had been applied and caused allergic inflammation in the nose. Furthermore, in the study by Egger C et al. [32] it was found that nasal exposure leading to allergic inflammation in the nose with a particular allergen molecule induced systemic rises of allergen-specific IgE levels to the allergen, which had caused allergic inflammation. Thus, results from these studies demonstrate that the measurement of allergen-specific IgE rises in serum is a biomarker for demonstrating that an allergic patient had a clinically relevant encounter to a particular allergen. In a pilot experiment, as shown in Figure 5, we demonstrated that the ISAC ImmunoCAP technology is suitable for detecting increases in grass pollen allergen-specific IgE levels in grass pollen-allergic patients after exposure to grass pollen. Significant increases in IgE levels specific to the major timothy grass pollen allergens Phl p 1, Phl p 2, Phl p 5, and Phl p 6 were detected approximately 3 months after exposure to grass pollen (Figure 5).

Furthermore, in order to identify which of the allergen molecules could be the possible trigger factor for a clinical reaction, we compared allergen-specific IgE levels in the preschool children at the time point of the acute wheezing attack and approximately three months later (Figure 3 and Figure 6). Alterations (i.e., increases; appearance of newly detectable IgE specificities) indicative of allergen contact were observed in 11 children (Figure 3 and Figure 6). Increases in allergen-specific IgE levels after the acute wheezing attack were noted in four children (i.e., #33, #90, #44, and #51) (Figure 3 and Figure 6). In two children (i.e., #51 and #44), increases to dog and horse allergens were found, whereas in the two other children (#33 and #90), food allergen-specific IgE increases were observed (Figure 3 and Figure 6). Newly detectable IgE sensitizations after the wheezing attacks that had not been detected in the sera collected during the acute visit were found in ten children (i.e., #33, #39, #47, #101, #16, #14, #30, #51, and #102) (Figure 3). There are two possibilities in regard to how they can be explained. First, it is possible that the newly detectable IgE sensitizations could not be measured in the first serum sample, because they were below the IgE levels that could be measured by the microarray technology. The second possibility is that the specific IgE was not yet detectable, because it was bound to IgE receptors on cells and only became detectable in serum once it had increased. This mechanism has been described for patients who had been treated with omalizumab, a monoclonal anti-IgE antibody that increased allergen-specific serum levels by blocking the binding of IgE to IgE receptors [33].

Allergen-specific IgE levels in the remaining IgE-sensitized children either remained unchanged, declined, or disappeared. Thus, we should note that increases in allergen-specific IgE levels and/or appearances of newly detectable IgE reactivities to allergens in follow-up serum samples for which no specific IgE could be measured during the first visit was interpreted as a sign of allergen contact that could have triggered the wheezing attack. It is, therefore, quite possible that allergen contact contributed to the wheezing attacks in 11 out of the 120 preschool children, either independently or in synergy with the viral infection [16]. Three IgE-sensitized children (i.e., #4, #50, and #104) had neither increases in RV-specific IgG antibodies nor alterations in IgE responses indicative of allergen contact, suggesting that their wheezing attack was neither due to an RV infection nor triggered by allergen exposure (Figure 3 and Figure 6).

## 4. Discussion

To the best of our knowledge, our study is the first to investigate, via microarray technology, the possible contribution of allergen exposure and rhinovirus infections as trigger factors for acute wheezing attacks in a cohort of preschool children. For this purpose, a meticulous assessment of molecular IgE sensitization profiles with a comprehensive panel of microarrayed allergen molecules was performed in the children, and we searched for alterations in allergen-specific IgE levels and RV-specific IgG responses, which could be indicative of allergen exposure and/or a RV infection in follow-up serum samples obtained from the children approximately 10 weeks after the wheezing attacks. Several results indicate that a period of approximately 10 weeks after allergen encounter or RV infection is suitable to measure increases of allergen-specific IgE levels and RV-specific IgG levels as biomarkers indicating a clinically relevant allergen exposure associated with symptoms of allergy and of a RV infection, respectively. Studies in which allergic patients have been exposed to defined allergen molecules causing respiratory symptoms have shown that the exposed patients showed specific IgE increases to the particular allergen molecule, which had caused allergic symptoms, which are most pronounced 5–10 weeks after allergen exposure [20,32]. The analysis of subjects who had been infected in a controlled RV exposure study indicated that species-specific rises of RV-specific IgG antibody levels can be detected 6 weeks after RV infection and the analysis in our study indicates that the increases of RV-specific IgG antibodies were even highest in sera from children obtained between >14 weeks after their wheezing attack when compared to sera obtained from children obtained at <10 weeks and 10–14 weeks (Figure 2).

It was demonstrated earlier that the MeDALL chip used for measuring allergen-specific IgE levels is more sensitive to the detection of IgE sensitizations in children than established methods of allergen extract-based skin testing, IgE serology with a panel of allergen extracts [34], or with combined serological tests for the detection of respiratory and food allergen-specific IgE [35]. The determination of the IgE reactivity profiles in the preschool children showed that the IgE sensitization rates to any of the microarrayed allergens varied from 14.3% in children aged 6–12 months to 29% in children aged 25–42 months, which was in agreement with the results obtained in birth cohort studies performed in Sweden [36,37]. Despite the relatively low rate of IgE sensitizations in the preschool children, we found evidence for allergen contact as a possible trigger factor for early childhood wheezing in approximately 10% of the children. In fact, we observed increases in allergen-specific IgE levels and *de novo* appearances of IgE specificities in follow-up serum samples of 11 out of the 120 children. Interestingly, increases in allergen-specific IgE levels to respiratory and food allergens were found, suggesting that allergen contact via the respiratory and gastrointestinal tract may have been associated with the wheezing attacks.

In the vast majority of children (i.e., in 87%; 104 of the 120 tested children), RV infections seemed to be the responsible trigger factors for the wheezing attack. Increases in RV-A-, RV-C-, and RV-B-specific IgG levels were found in 34%, 27%, and 19%, respectively, of the follow-up serum samples. In 6% of the sera, increases in IgG against at least two different RV-species were found, which may be caused by mixed infections by different RV strains. In sixteen of the 120 children (i.e., 13%), we did not detect relevant increases in RV-specific IgG levels in the follow-up sera or alterations in allergen-specific IgE indicative of allergen exposures. It is, therefore, quite possible that the wheezing attacks in these children were triggered by other factors, for example, by infections with other respiratory viruses such as RSV [7] or by exposure to irritants such as tobacco smoke [38].

One finding deserving attention was that increases in RV-specific IgG responses indicative of RV-triggered wheezing were found equally distributed in IgE-sensitized and in non-IgE-sensitized children. This indicates that atopy and/or allergic sensitization is not a prerequisite for the occurrence of RV-induced wheezing attacks in preschool children [39].

One limitation of our study is that it was performed as a retrospective analysis of sera from children having experienced a wheezing attack, but for ethical reasons, it was not possible to perform a prospective study in which children were exposed experimentally to RV and/or allergens for the induction of wheezing. Accordingly, we cannot exclude the occurrence of infections with additional RV strains in the time window between the first and second blood sampling and, thus, there is no definitive proof that RV infections and allergen contact were indeed responsible for the wheezing attacks. However, it should be mentioned that the children investigated in our study did not have exacerbations of respiratory disease in the time window until the follow-up visit so that an intervening allergen or RV exposure capable of triggering a wheezing attack in the follow up period is very unlikely. Another limitation of our study is the lack of a non-wheezing control group. However, the fact that increases in RV-specific antibodies were not detected in any of the tested children indicates the specificity of the microarray. In addition, we already demonstrated in an earlier exposure study that the magnitudes of increases in RV-specific antibodies were related with the severity of respiratory symptoms; however, the cut-off levels for RV-specific IgG increases related to respiratory disease exacerbations have not yet been established [22].

Evidence that RV infections are trigger factors for wheezing attacks has, thus far, mainly been obtained by demonstrating the presence of RV-derived nucleic acid in children at the time of the wheezing attack. In our cohort, we identified, in 108 of the 120 children, RV-derived nucleic acid by nested PCR targeting the VP4/VP2 region and subsequent sequencing [24], which is in good agreement with our serological data. However, there are at least two advantages of using serological assays in studies investigating the role of RV infections as possible trigger factors for wheezing/asthma attacks over the standard PCR-based testing. First, serology demonstrates that an RV infection has indeed occurred and has affected the immune response. In this context, we should mention that RV RNA was detected in large numbers of asymptomatic infants and children, and, accordingly, it is not a specific parameter for a RV-triggered wheezing attack [40,41,42,43]. Second, the use of micro-arrayed peptides from RV strains allows species-specific diagnoses [24]. Now that both types of assays, the nucleic acid detection and the measurement of RV-group-specific antibodies in sera of convalescent subjects are available, both types of assays can be compared in further studies. Our study showed good concordance of PCR-based and serological tests.

Thus, our study demonstrates that microarrays containing comprehensive sets of allergens, together with microarrays comprising the epitopes of common RV-strains, may be useful for identifying allergen-specific IgE and/or RV infections as possible trigger factors for wheezing attacks in preschool children, which may have implications in asthma prediction and prevention [44]. Further studies investigating the effects of personalized treatment schedules, taking into account the trigger factors responsible for the wheezing attacks, are warranted.

## Figures and Tables

**Figure 1 viruses-13-00915-f001:**
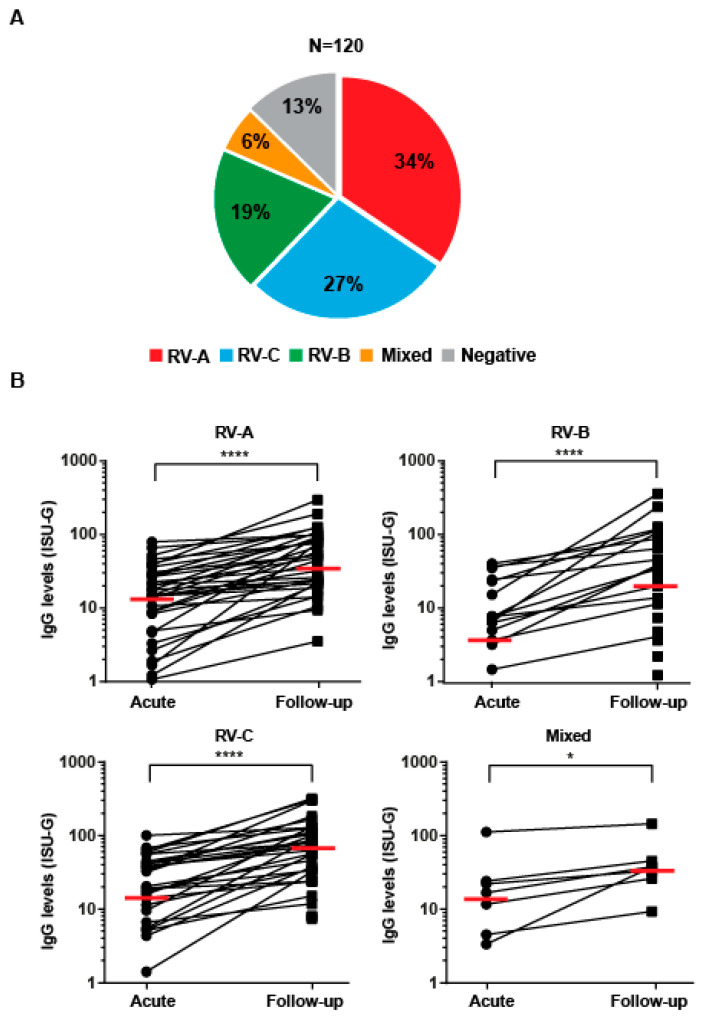
Increases in RV species-specific antibody responses in preschool children presenting with acute wheeze. (**A**) Pie chart showing percentages of children (*n* = 120) with increases in IgG antibodies specific to VP1-derived peptides representing the three RV species (red: RV-A; blue: RV-C; green: RV-B; yellow: mixed RV species; grey: no RV-specific responses). (**B**) Levels of RV-specific IgG antibodies (y-axes: ISU-G) measured during the acute and the follow-up visits (x-axes) in children for which RV-A (*n* = 41), RV-B (*n* = 23), RV-C (*n* = 33), and mixed RV (*n* = 7) were identified as culprit RV species. IgG responses to the peptide with the largest IgG increase are shown, where horizontal bars represent median values of RV-specific IgG levels. Statistically significant differences between subjects were calculated via the Wilcoxon matched-pairs signed rank test, and they are indicated on the graphs (**** *p* < 0.0001; * *p* < 0.05).

**Figure 2 viruses-13-00915-f002:**
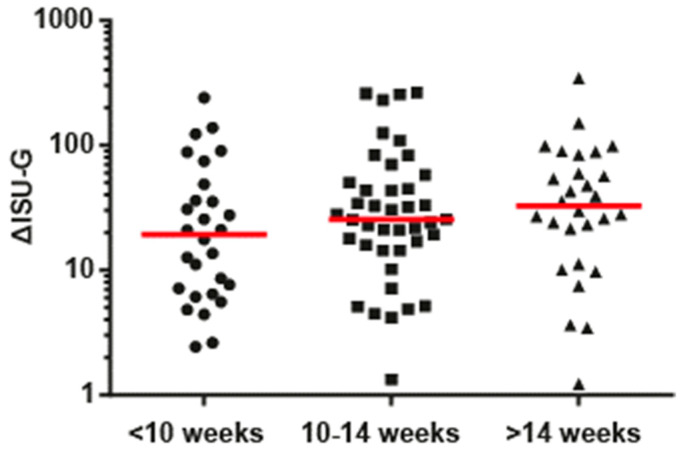
Increases of RV-specific IgG responses in children according to the time points of the follow-up visits. Shown are increases of IgG responses to VP1-derived peptides of RV-A, -B, and -C (*y*-axis: ΔISU-G) measured in serum samples collected at the day of a wheezing attack and afterwards. Children were grouped according to the time points of the follow-up visits (*x*–axis: <10 weeks, between 10–14 weeks, and >14 weeks). Horizontal lines represent median values of increases of RV-specific IgG levels.

**Figure 3 viruses-13-00915-f003:**
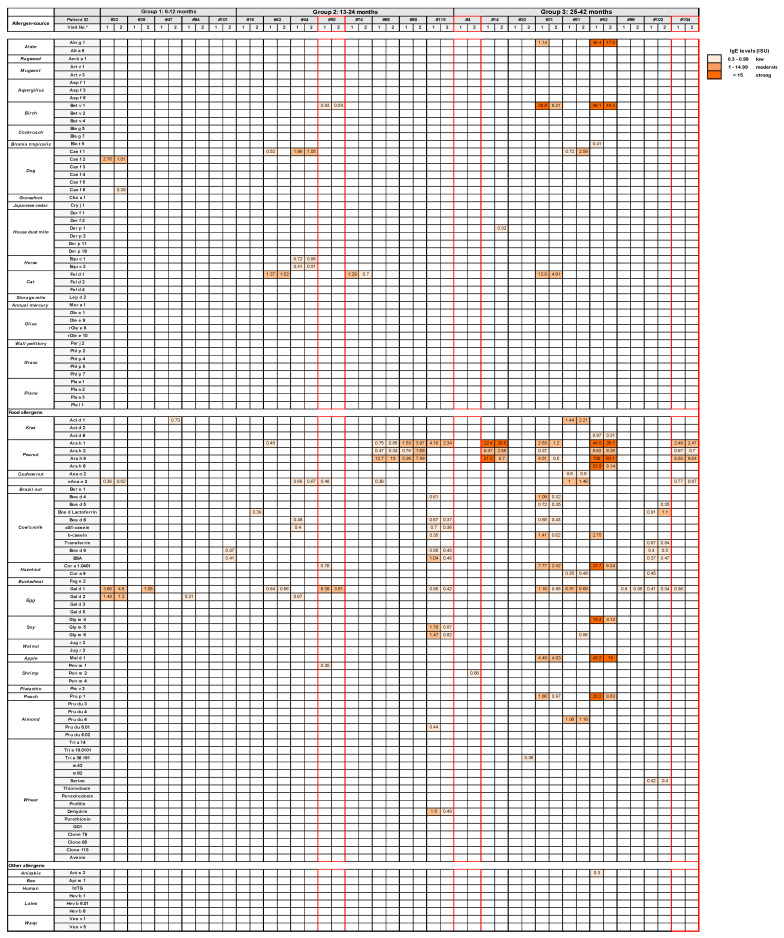
IgE levels to each of the individual allergen molecules on the MeDALL microarray. Specific IgE levels (in ISU-E) are displayed for each allergen molecule (left margin) at the acute (1) and follow-up visits (2) (white: no IgE sensitization, IgE < 0.3 ISU-E; light orange: = or >0.3–1 ISU-E; medium orange >1–15 ISU-E; dark orange >15 ISU-E for IgE-sensitized children from the three age groups, as indicated on top. Children without increases in RV-specific IgG antibodies are outlined in red.

**Figure 4 viruses-13-00915-f004:**
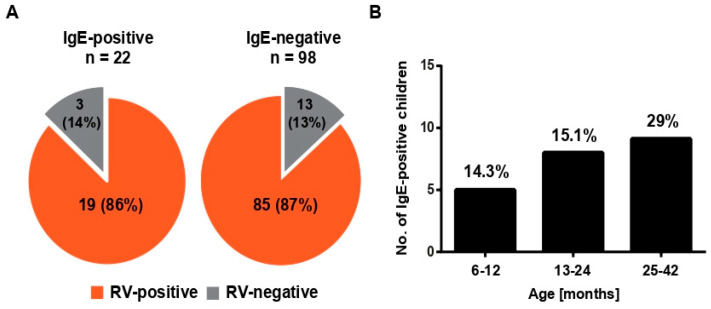
Allergen-specific IgE responses in preschool children presenting with acute wheeze. (**A**) Percentages of wheezing children with and without increases in RV-specific IgG among IgE-sensitized (*n* = 22, left) and non-sensitized (*n* = 98, right) children; (**B**) numbers (*y*-axis) and percentages of IgE-sensitized children in the three age groups.

**Figure 5 viruses-13-00915-f005:**
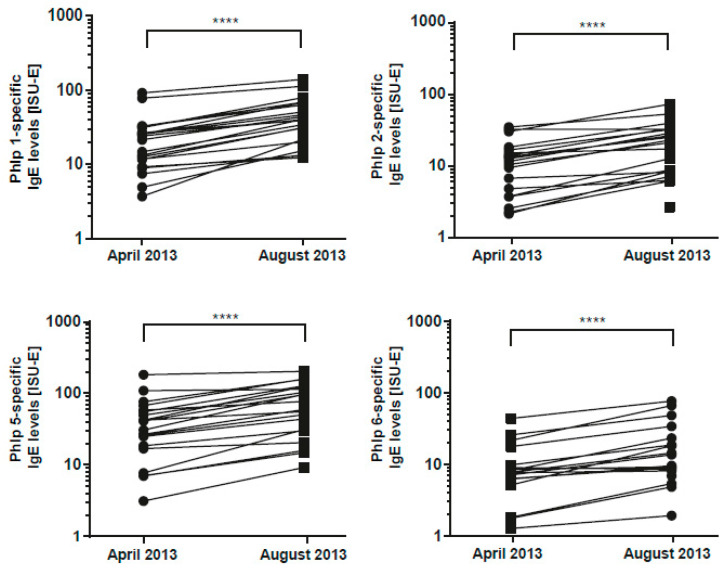
Grass pollen allergen-specific IgE levels before and after exposure to grass pollen. Graphs show IgE levels specific to the 4 major grass pollen allergens (Phl p 1, Phl p 2, Phl p 5, and Phl p 6) in grass pollen-allergic patients (*n* = 21) measured before (April 2013) and after (August 2013) exposure to seasonal grass pollen. Differences between IgE levels were calculated via the Wilcoxon matched-pairs signed rank test, and p values are indicated (**** *p* < 0.0001).

**Figure 6 viruses-13-00915-f006:**
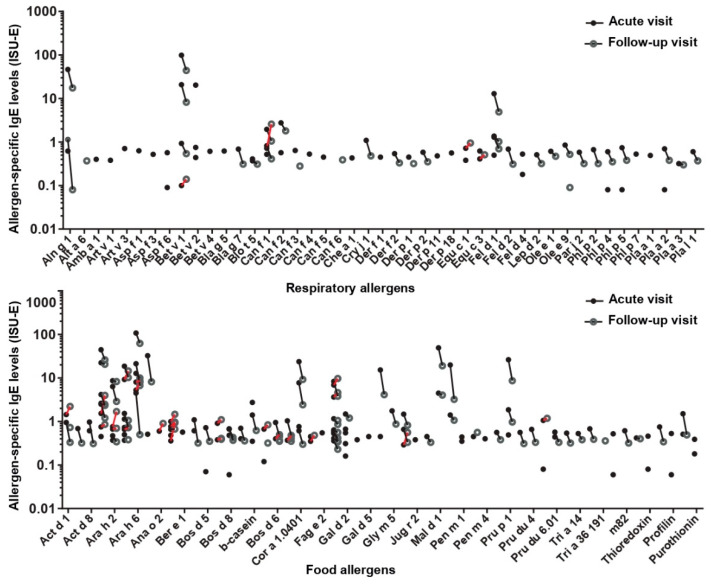
Alterations in allergen-specific IgE levels in sera during the acute and follow-up visits. Graphs show specific IgE levels (y-axes: ISU-IgE) for respiratory (**top**) and food allergens (**bottom**), determined during the acute and follow-up visits. Increases are indicated in red.

**Table 1 viruses-13-00915-t001:** Characterization of the analyzed wheezing children.

Characteristics	Value
**Age in months, median (min-max)**	**18 (6–42)**
6–12 months, *n* = 35 (median)	10
13–24 months, *n* = 53 (median)	17
25–42 months, *n* = 32 (median)	32
**Male, n (%)**	**76 (63)**
6–12 months, *n* (%)	35 (29)
13–24 months, *n* (%)	53 (44)
25–42 months, *n* (%)	31 (26)
**Ever wheeze before, *n* (%)**	**92 (77)**
**Allergic sensitization ^1^, *n* (%)**	**22 (18)**
6–12 months, *n* (%)	5 (4.2)
13–24 months, *n* (%)	8 (6.6)
25–42 months, *n* (%)	9 (7.5)
**Weeks until follow-up, median (min-max)**	**11 (6–32)**
6–12 months, median (min-max)	11 (7–27)
13–24 months, median (min-max)	11 (7–24)
25–42 months, median (min-max)	12.5 (9–30)

^1^ Allergen-specific IgE ≥ 0.3 ISU-E to at least one allergen measured by MeDALL allergen chip.

## Data Availability

Primary data that support the findings of this study are available from the corresponding author on reasonable request.

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
