# Peer review of "Microarray Technology May Reveal the Contribution of Allergen Exposure and Rhinovirus Infections as Possible Triggers for Acute Wheezing Attacks in Preschool Children"

_viruses, 2021, doi:10.3390/v13050915_

Round 1
Reviewer 1 Report
General Comments:
This manuscript describes an effort to assess the contribution of allergic sensitization and/or rhinovirus infection to wheezing episodes in infants. This issue has been addressed in a number of studies including some with prospective, longitudinal designs. Unfortunately, the design of this study, based on serologic assays, does little to advance our existing understanding of this problem.
Given the rapidity of IgE responses to allergen challenge it is not clear why evaluation of “convalescent” sera was included in the assessment of allergic sensitization. It seems likely that relevant sensitization would be detected at the time of the wheezing illness.
The evaluation of rhinovirus infection is also problematic. Serologic assays may have a role in the assessment of rhinovirus infection but specific diagnosis with PCR assays at the time of infection is readily available and would be much preferred. The use of a panel of VP1 antigens with an interval of as long as 30 weeks (median 11 weeks) before collection of the convalescent serum raises the possibility that that the detected infection could be unrelated to the wheezing episode. The finding that 87% of patients had a response on the RV panel with only 30 serotypes tested is higher than expected and also suggests that infections unrelated to the wheezing episode may have been included. The advantage of using serology is not obvious; if the authors have a convincing argument that should be included in the manuscript.
Comparison of the serologic responses to rhinovirus and the allergens in a non-wheezing control group would perhaps be beneficial to the interpretation of this study.
Specific Comments:
Line 102: Little information about the research subjects is included. Were any virologic assays done for detection of infection at the time of the wheezing diagnosis?
Line 151: This table is mostly blank.
Line 162: The data provided in this paragraph seem mostly meaningless. Diagnostic serologic responses to rhinovirus infection depend on serotype-specific antibody increases. It is not clear what can be learned by a description of aggregate antibody rises to rhinovirus-species.
Line 199: Figure 2 is quite large and difficult to follow (especially since it appears to be written in Greek). Would perhaps be suitable for inclusion in the supplemental material if translated.
Line 214: It is not clear how evidence for allergen exposure after the acute wheezing episode can be used as an indication that sensitization is a trigger for the wheezing.
Author Response
Response to Reviewer 1:
Comments and Suggestions for Authors
General Comments:
This manuscript describes an effort to assess the contribution of allergic sensitization and/or rhinovirus infection to wheezing episodes in infants. This issue has been addressed in a number of studies including some with prospective, longitudinal designs. Unfortunately, the design of this study, based on serologic assays, does little to advance our existing understanding of this problem.
Response: We thank the reviewer for the comment but must respectfully disagree with the statement. We have actually quoted some of the prospective longitudinal studies studying the possible contribution of allergic sensitization and rhinovirus infections to wheezing episodes infants. However, our study is the first to relate increases of allergen-specific IgE levels and rhinovirus-specific antibodies measured with defined allergen molecules and defined peptides from RV coat proteins as indicators for specific immune responses in children with documented wheezing episodes. In order to conduct such an analysis serum samples must be obtained at the time point when the wheezing episode occurs in a particular child and some weeks later in order to be able to assess if allergen contact and/or an RV infection has boosted the specific antibody response. To the best of our knowledge the question whether allergen contact and/or RV infection has boosted specific antibody levels in the context of a wheezing attack has never been investigated before. We disagree with the reviewer that the analysis of this question does little to advance our existing understanding of the problem because only the analysis of the effects of allergen exposure and/or RV infections on the specific immune response allows to make conclusions if allergen and/or RV contact had an effect on the immune system of the child during the wheezing episode. We have revised our manuscript to clarify this misunderstanding (see lines 102-108 of the revised manuscript).
Given the rapidity of IgE responses to allergen challenge it is not clear why evaluation of “convalescent” sera was included in the assessment of allergic sensitization. It seems likely that relevant sensitization would be detected at the time of the wheezing illness.
Response: Our study is a real-life study and not an allergen challenge study. The children in our study presented at the clinic with an acute wheezing attack which was immediately treated. Accordingly, a retrospective analysis of the boosts of specific IgE and RV-specific antibodies was performed to analyse the possible culprit trigger factor. Our results also show that even if one would have determined the complete IgE sensitization profile at the time of the first visit one would not have known what particular allergen might have triggered the wheezing attack. A retrospective identification of a particular allergen molecule as possible trigger factor for a clinical reaction can be assessed by the boost of the allergen-specific IgE levels in a follow up serum sample obtained some weeks later as shown in Figure S1 of our manuscript which upon request of another reviewer has now become Figure 4 of the revised manuscript. We have revised our manuscript to explain the design of the serological analysis and quoted earlier studies, which have demonstrated that allergen contact not only causes a clinical reaction but also leads to boosts of allergen-specific IgE responses, which can be measured in “convalescent sera” (see lines 246-247 and 261-262 of the revised manuscript).
The evaluation of rhinovirus infection is also problematic. Serologic assays may have a role in the assessment of rhinovirus infection but specific diagnosis with PCR assays at the time of infection is readily available and would be much preferred. The use of a panel of VP1 antigens with an interval of as long as 30 weeks (median 11 weeks) before collection of the convalescent serum raises the possibility that that the detected infection could be unrelated to the wheezing episode. The finding that 87% of patients had a response on the RV panel with only 30 serotypes tested is higher than expected and also suggests that infections unrelated to the wheezing episode may have been included. The advantage of using serology is not obvious; if the authors have a convincing argument that should be included in the manuscript.
Response: We agree with the reviewer that PCR tests can demonstrate the presence of virus-derived nucleic acid but these tests do not necessarily prove that the detected virus had caused an infection and was responsible for clinical symptoms (see: Pavia, A. T. Viral infections of the lower respiratory tract: old viruses, new viruses, and the role of diagnosis. Clin. Infect. Dis. 52, 284–289 (2011)). It should be also mentioned that RV RNA was detected in large numbers of asymptomatic infants and children (Advani, S., Sengupta, A., Forman, M., Valsamakis, A. & Milstone, A. M. Detecting respiratory viruses in asymptomatic children. Pediatr. Infect. Dis. J. 31, 1221–1226 (2012); van der Zalm, M. M. et al. Respiratory pathogens in children with and without respiratory symptoms. J. Pediatr. 154, 396–400 (2009); Winther, B., Hayden, F. G. & Hendley, J. O. Picornavirus infections in children diagnosed by RT-PCR during longitudinal surveillance with weekly sampling: association with symptomatic illness and effect of season. J. Med. Virol. 78, 644–650 (2006)). There are at least two advantages of measuring increases of RV-specific antibodies over PCR testing. First, serology demonstrates that an RV infection has indeed occurred and affected the immune response. Second, the use of micro-arrayed peptides from RV strains allows species-specific diagnosis. We have mentioned these considerations in our revised manuscript but also mentioned as a possible limitation of our study, that results obtained for samples collected longer than 10 weeks (i.e., the optimal time for measuring increases of RV-specific antibody responses, see: Niespodziana K, Cabauatan CR, Jackson DJ, et al. Rhinovirus-induced VP1-specific antibodies are group-specific and associated with severity of respiratory symptoms. EBioMedicine 2014;2(1):64-70) after the wheezing attack maybe due to a later infection not related to the wheezing episode (see lines 338-340 and 349-360) of the revised manuscript).
Comparison of the serologic responses to rhinovirus and the allergens in a non-wheezing control group would perhaps be beneficial to the interpretation of this study.
Response: We thank the reviewer for this comment. Unfortunately, such a group was not available and we have mentioned this as a possible limitation of our study (see lines 358-359 of the revised manuscript). However, the fact, that increases of RV-specific antibodies were not detected in all tested children indicates the specificity of the microarray. Also we have shown in an earlier exposure study that the magnitudes of increases of RV-specific antibodies are related with the severity of respiratory symptoms (Niespodziana K, Cabauatan CR, Jackson DJ, et al. Rhinovirus-induced VP1-specific antibodies are group-specific and associated with severity of respiratory symptoms. EBioMedicine 2014;2(1):64-70). These considerations were also added to the revised manuscript (see lines 341-348).
Specific Comments:
Line 102: Little information about the research subjects is included. Were any virologic assays done for detection of infection at the time of the wheezing diagnosis?
Response: Following the reviewer’s suggestion we included now the information on virologic assays in the revised version of the manuscript. It should be noted that PCR-based evidence for RV infections was obtained for the majority of children who showed increases of RV-specific IgG levels by serology (see lines 117-124).
Line 151: This table is mostly blank.
Response: We are very sorry that the reviewer received a blank table. We have now included the complete Table I as a picture in the revised manuscript (see lines 174-175).
Line 162: The data provided in this paragraph seem mostly meaningless. Diagnostic serologic responses to rhinovirus infection depend on serotype-specific antibody increases. It is not clear what can be learned by a description of aggregate antibody rises to rhinovirus-species.
Response: We must respectfully disagree with the reviewer. It has been shown that cumulative RV-specific antibody responses are useful parameters (see: Megremis S, Niespodziana K, Cabauatan C, et al. Rhinovirus species-specific antibodies differentially reflect clinical outcomes in health and asthma. Am J Respir Crit Care Med 2018;198(12):1490-1499). In fact, the microarray used in this study contained 30 N-terminal VP1 peptides including 18 peptides of RV-A species, 9 peptides of RV-B and 3 peptides of RV-C species. Due to the fact, that peptides from RV strains belonging to the same RV species show higher sequence similarity than those from unrelated species, cross-reactivity to peptides from one species can be observed to establish cumulative species-specific antibody levels. This was explained in the revised manuscript (see lines 164-168).
Line 199: Figure 2 is quite large and difficult to follow (especially since it appears to be written in Greek). Would perhaps be suitable for inclusion in the supplemental material if translated.
Response: We feel sorry that the reviewer received Figure 2 in a low quality. Unfortunately, we were not aware about that the submitted manuscript was converted in an unusual format. We have replaced the figure with a new Figure 2 and suggest to leave it in the main manuscript because it reveals the details of the findings (see lines 221-222).
Line 214: It is not clear how evidence for allergen exposure after the acute wheezing episode can be used as an indication that sensitization is a trigger for the wheezing.
Response: We thank the reviewer for this comment and refer to our above response “A retrospective identification of a particular allergen molecule as possible trigger factor for a clinical reaction can be assessed by the boost of the allergen-specific IgE levels in a follow up serum sample obtained some weeks later as shown in Figure S1 of our manuscript. We have revised our manuscript to explain the design of the serological analysis and quoted earlier studies which have demonstrated that allergen contact not only causes a clinical reaction but also leads to boosts of allergen-specific IgE responses which can be measured in “convalescent sera” (see lines 246-2247 and 261-262 of the revised manuscript).
Reviewer 2 Report
- Suggestions to Author/s
- Dear Dr. RudolfValenta, as a selected reviewer I made a prompt check of Your article:”Microarray technology may reveal contribution of allergen exposure and rhinovirus infections as possible triggers for acute wheezing attacks in preschool children” and found it: (x) Excellent, accept the submission (5) . So I will suggest to the editor, to include it into the first available journal’s number.
2. In Your article I found some minor mistakes, for which I kindly ask You to correct them in a following way:
Line No: Correction ( Delete/Add):
------------------------------------------------
27 Del.: has been Add: was
49 Del.: 3 Add: three
53 Del.: has been Add: was
56 Del.: 3 Add: three
67 Del.: has been Add: was
68 Add: provoke the; Add: of the
69 Add: , Add: on
74 Del.: have been Add: were
75 Del.: has been Add: was Add: the
81 Del.: has been Add: was
86 Del.: has been Add: was
130 Add: s Add: s
133 Add: of the
Table 1: Correct the table with the real values
217 Del.: has been Add: was
247 Add: for
254 Del.: has been Add: was
274 Del.: have been Add: be

Author Response
Response to Reviewer 2:
Suggestions to Author/s
Dear Dr. Rudolf Valenta, as a selected reviewer I made a prompt check of Your article: ”Microarray technology may reveal contribution of allergen exposure and rhinovirus infections as possible triggers for acute wheezing attacks in preschool children” and found it: (x) Excellent, accept the submission (5) . So I will suggest to the editor, to include it into the first available journal’s number.
Response: We thank this reviewer for the positive comments regarding our work.
- In Your article I found some minor mistakes, for which I kindly ask You to correct them in a following way:
Line No: Correction (Delete/Add):
------------------------------------------------
27 Del.: has been Add: was
49 Del.: 3 Add: three
53 Del.: has been Add: was
56 Del.: 3 Add: three
67 Del.: has been Add: was
68 Add: provoke the; Add: of the
69 Add: , Add: on
74 Del.: have been Add: were
75 Del.: has been Add: was Add: the
81 Del.: has been Add: was
86 Del.: has been Add: was
130 Add: s Add: s
133 Add: of the
Table 1: Correct the table with the real values
217 Del.: has been Add: was
247 Add: for
254 Del.: has been Add: was
274 Del.: have been Add: be
Response: Following reviewer’s suggestions, we have included all of the requested changes in the revised manuscript (see lines 27, 50, 54, 58, 68, 69, 71, 75, 76, 82, 87, 148, 151, 297, 304, 323).
Reviewer 3 Report
- Allergen exposure and rhinovirus infections are triggers for wheezing in childhood. Allergic sensitization and RV infection can act synergistically to promote the development of asthma. RV infection leads to increases of virus specific IgG.
- They analysed serum samples of 120 children who had been admitted to hospital due to acute symptoms. They determined the antibody reactivity profile by MeDALL allergen and PeDicta RV chip. They determined allergen-specific IgE and RV-specific responses. Their overall strategy seem like a good one. I am not able to judge whether there are major problems with their used methods. I would be very happy to see a short section where the authors give some insights into the pitfalls of their techniques. I do not see other experiments that would greatly improve the quality of the manuscript.
- They found RV specific increases of IgG levels in most of the children and allergen-specific IgE. Unfortunately, figure 2 can not be read and is therefore not understandable. They also found no evidence that RV-induced wheeze is more common in IgE-sensitized children as compared to non-IgE-sensitized children. As the description of the results often refers to Figure 2 and I cannot read that figure, it was very difficult to follow the conclusions. The authors describe in line 232/233 that new IgE sensitizations were detected in 11 children and they conclude that allergen exposure has contributed to the wheezing attacks. I cannot follow this argumentation. If children were non sensitized to the time point of wheezing due to the RV infection, how could allergen exposure have contributed to the wheezing? Before IgE is manifested in the body, it takes up to 10 days for the first phase of sensitization, and more than one exposure. The sensitization phase to an allergen is symptom-free. For me, it sounds as if the RV infection (due to tissue injury) paved the way for sensitization before manifestation of hypersensitivity.
- Nevertheless, the results are interesting and important and other researchers will be interested in reading the study.
- The way how the manuscript fits together could be improved. I found no clear description of the hypothesis. I am not sure what research question did the authors address. I think, there is good argument for why this question is important but I could not find it. That is to related to the two sentences 84/85 and 86/87 which are not easy to understand. So, the manuscript is not entirely easy to read. I also would recommend to clean phrases to that they are used correctly. For example, the authors measured allergen-specific IgE and not allergen exposure. The measurement of the IgEs is only an indirect approach for the allergen exposure.
Author Response
Response to Reviewer 3:
Allergen exposure and rhinovirus infections are triggers for wheezing in childhood. Allergic sensitization and RV infection can act synergistically to promote the development of asthma. RV infection leads to increases of virus specific IgG. They analysed serum samples of 120 children who had been admitted to hospital due to acute symptoms. They determined the antibody reactivity profile by MeDALL allergen and PreDicta RV chip. They determined allergen-specific IgE and RV-specific responses. Their overall strategy seem like a good one. I am not able to judge whether there are major problems with their used methods. I would be very happy to see a short section where the authors give some insights into the pitfalls of their techniques. I do not see other experiments that would greatly improve the quality of the manuscript.
Response: We thank the reviewer for these positive comments. Following the comments of the other reviewers we have explained the applied methods with their strengths and possible limitations (see lines 349-360 of the revised manuscript).
They found RV specific increases of IgG levels in most of the children and allergen-specific IgE. Unfortunately, figure 2 can not be read and is therefore not understandable.
Response: We apologize that the reviewer received Figure 2 in a low quality. This must have happened during the conversion of the submitted Figure by the electronic system. We were not aware about that the submitted manuscript will be placed in a special format. We have replaced the figure with a new Figure 2 (see lines 221-222 and the attachment).
They also found no evidence that RV-induced wheeze is more common in IgE-sensitized children as compared to non-IgE-sensitized children. As the description of the results often refers to Figure 2 and I cannot read that figure, it was very difficult to follow the conclusions. The authors describe in line 232/233 that new IgE sensitizations were detected in 11 children and they conclude that allergen exposure has contributed to the wheezing attacks. I cannot follow this argumentation. If children were non sensitized to the time point of wheezing due to the RV infection, how could allergen exposure have contributed to the wheezing? Before IgE is manifested in the body, it takes up to 10 days for the first phase of sensitization, and more than one exposure. The sensitization phase to an allergen is symptom-free. For me, it sounds as if the RV infection (due to tissue injury) paved the way for sensitization before manifestation of hypersensitivity.
Response: We agree that Figure 2 is needed to follow the description and provided a new, hopefully well readable version of it in the revised manuscript. The comment of the reviewer about “new sensitizations” is excellent. We have revised this definition and replaced in by “newly detectable IgE sensitizations”. There are two possibilities how they can be explained. First, it is possible that the newly detectable IgE sensitizations were not detected because they were below what can be measured by the microarray technology. We think that this is less likely because the microarray technology was found to be equally sensitive than the quantitative ImmunoCAP technology (see: Advances in allergen-microarray technology for diagnosis and monitoring of allergy: the MeDALL allergen-chip. Lupinek C, Wollmann E, Baar A, Banerjee S, Breiteneder H, Broecker BM, Bublin M, Curin M, Flicker S, Garmatiuk T, Hochwallner H, Mittermann I, Pahr S, Resch Y, Roux KH, Srinivasan B, Stentzel S, Vrtala S, Willison LN, Wickman M, Lødrup-Carlsen KC, Antó JM, Bousquet J, Bachert C, Ebner D, Schlederer T, Harwanegg C, Valenta R. Methods. 2014 Mar 1;66(1):106-19. doi: 10.1016/j.ymeth.2013.10.008. Epub 2013 Oct 22). The second possibility is that specific IgE was not yet detectable because it was bound to IgE receptors on cells and only became detectable in serum once it increased. We have own unpublished data supporting such a mechanism and there is also a study describing that the administration of the anti-IgE antibody omalizumab which prevents IgE binding to IgE receptors makes IgE sensitizations detectable by serology (see: Influence of Omalizumab on allergen-specific IgE in patients with adult asthma. Mizuma H, Tanaka A, Uchida Y, Fujiwara A, Manabe R, Furukawa H, Kuwahara N, Fukuda Y, Kimura T, Jinno M, Ohta S, Yamamoto M, Matsukura S, Adachi M, Sagara H. Int Arch Allergy Immunol. 2015;168(3):165-72. doi: 10.1159/000442668. Epub 2016 Jan 21). We revised the manuscript to include the two explanations (see lines 272-279 of the revised manuscript) and thank the reviewer for having brought up this important issue.
Nevertheless, the results are interesting and important and other researchers will be interested in reading the study. The way how the manuscript fits together could be improved. I found no clear description of the hypothesis. I am not sure what research question did the authors address. I think, there is good argument for why this question is important but I could not find it. That is to related to the two sentences 84/85 and 86/87 which are not easy to understand. So, the manuscript is not entirely easy to read. I also would recommend to clean phrases to that they are used correctly. For example, the authors measured allergen-specific IgE and not allergen exposure. The measurement of the IgEs is only an indirect approach for the allergen exposure.
Response: Following the reviewer’s suggestion and comments of the other reviewers we included a more clear description of the hypothesis (see lines 95-101) in the revised manuscript. We also used the term allergen-specific IgE increases as sign of allergen contact instead of allergen exposure where appropriate (see lines 265, 285, 288, 312, 317, 341, 363 of the revised manuscript).
Reviewer 4 Report
Acute wheezing has been attributed to rhinovirus infection and/or exposure of sensitized individuals to allergens. Infection and allergen exposure have been associated with increased antibody titers several weeks after the acute wheezing episode. In this study, allergen and RV-VP1 peptide arrays are used to examine a cohort of 120 children (6- 42 months) at the time of acute wheezing and 6-30 weeks later.
Analysis of RV revealed that 87% of children showed an increase in IgG titers for one of the RV peptides on the chip. In contrast, only 18% of children demonstrated sensitization at the time of the wheezing episode to any of the 160 common allergens on the MeDALL chip, with prevalence trending upward with increased age. There was no difference in the percentage of IgE positive and negative children who were positive for RV infection. Subsequent analysis of IgE levels weeks after the wheezing incident identified 11 children with increased IgE titers, indicating possible allergen exposure at the time of the respiratory exacerbation.
This straight forward study demonstrates the potential utility of allergen/viral arrays for post-exposure identification of the causes of acute wheezing episodes in children and perhaps other patient populations. The data is convincing and generally support the conclusions of the manuscript. However, a few details were missing that need to be added before a complete review can occur. First, Table 1 is missing most of the data (except for the ages of the participants), also it’s not clear what the two columns represent. Second, Figure 2 is essentially uninterpretable; after enlarging it the image became pixelated before I was able to read any details in it. Lastly, the authors state that the follow up IgE analysis occurred ‘…when no sign of infection was present’ but don’t provide any information on how this was assessed (or why it was important). Unless there are space constraints, Figure S1 should be incorporated into the manuscript.
The last paragraph of the results section was confusing and needs better clarification of how exposure vs sensitization are being used.
Author Response
Response to Reviewer 4:
Acute wheezing has been attributed to rhinovirus infection and/or exposure of sensitized individuals to allergens. Infection and allergen exposure have been associated with increased antibody titers several weeks after the acute wheezing episode. In this study, allergen and RV-VP1 peptide arrays are used to examine a cohort of 120 children (6-42 months) at the time of acute wheezing and 6-30 weeks later.Analysis of RV revealed that 87% of children showed an increase in IgG titers for one of the RV peptides on the chip. In contrast, only 18% of children demonstrated sensitization at the time of the wheezing episode to any of the 160 common allergens on the MeDALL chip, with prevalence trending upward with increased age. There was no difference in the percentage of IgE positive and negative children who were positive for RV infection. Subsequent analysis of IgE levels weeks after the wheezing incident identified 11 children with increased IgE titers, indicating possible allergen exposure at the time of the respiratory exacerbation.This straight forward study demonstrates the potential utility of allergen/viral arrays for post-exposure identification of the causes of acute wheezing episodes in children and perhaps other patient populations. The data is convincing and generally support the conclusions of the manuscript. However, a few details were missing that need to be added before a complete review can occur. First, Table 1 is missing most of the data (except for the ages of the participants), also it’s not clear what the two columns represent. Second, Figure 2 is essentially uninterpretable; after enlarging it the image became pixelated before I was able to read any details in it. Lastly, the authors state that the follow up IgE analysis occurred ‘...when no sign of infection was present’ but don’t provide any information on how this was assessed (or why it was important). Unless there are space constraints, Figure S1 should be incorporated into the manuscript. The last paragraph of the results section was confusing and needs better clarification of how exposure vs sensitization are being used.
Response: We thank the reviewer for the careful evaluation of our work and apologize for the bad quality of Table I and Figure 2 which seems to have occurred during the conversion of the paper in a new format during submission. We, therefore, replaced them with a new Table I and Figure 2 in the revised manuscript. Following reviewer’s suggestion we incorporated Figure S1 as a new Figure 4 in the revised manuscript (see line 252-260). We also added information how we assessed that there were no signs of infection and why this was important (see lines 242-245). Finally, we have re-written the last paragraph of the results section for a better clarification of how the terms exposure vs sensitization were used (see lines 281-285).
Round 2
Reviewer 1 Report
General Comments:
This manuscript describes an effort to assess the contribution of allergic sensitization and/or rhinovirus infection to wheezing episodes in infants. This issue has been addressed in a number of studies including some with prospective, longitudinal designs. Unfortunately, the design of this study, based on serologic assays, does little to advance our existing understanding of this problem.
Response: We thank the reviewer for the comment but must respectfully disagree with the statement. We have actually quoted some of the prospective longitudinal studies studying the possible contribution of allergic sensitization and rhinovirus infections to wheezing episodes infants. However, our study is the first to relate increases of allergen-specific IgE levels and rhinovirus-specific antibodies measured with defined allergen molecules and defined peptides from RV coat proteins as indicators for specific immune responses in children with documented wheezing episodes. In order to conduct such an analysis serum samples must be obtained at the time point when the wheezing episode occurs in a particular child and some weeks later in order to be able to assess if allergen contact and/or an RV infection has boosted the specific antibody response. To the best of our knowledge the question whether allergen contact and/or RV infection has boosted specific antibody levels in the context of a wheezing attack has never been investigated before. We disagree with the reviewer that the analysis of this question does little to advance our existing understanding of the problem because only the analysis of the effects of allergen exposure and/or RV infections on the specific immune response allows to make conclusions if allergen and/or RV contact had an effect on the immune system of the child during the wheezing episode. We have revised our manuscript to clarify this misunderstanding (see lines 102-108 of the revised manuscript).
The authors’ do not provide a convincing argument that this study provides information that is not provided by studies that assess allergen-specific IgE concentrations and RV detection by PCR. Why a rise in IgE antibody after the fact of the wheezing episode should provide additional information is not clear. Rises in non-serotype specific antibody contribute no information about the RV infection.
Given the rapidity of IgE responses to allergen challenge it is not clear why evaluation of “convalescent” sera was included in the assessment of allergic sensitization. It seems likely that relevant sensitization would be detected at the time of the wheezing illness.
Response: Our study is a real-life study and not an allergen challenge study. The children in our study presented at the clinic with an acute wheezing attack which was immediately treated. Accordingly, a retrospective analysis of the boosts of specific IgE and RV-specific antibodies was performed to analyse the possible culprit trigger factor. Our results also show that even if one would have determined the complete IgE sensitization profile at the time of the first visit one would not have known what particular allergen might have triggered the wheezing attack. A retrospective identification of a particular allergen molecule as possible trigger factor for a clinical reaction can be assessed by the boost of the allergen-specific IgE levels in a follow up serum sample obtained some weeks later as shown in Figure S1 of our manuscript which upon request of another reviewer has now become Figure 4 of the revised manuscript. We have revised our manuscript to explain the design of the serological analysis and quoted earlier studies, which have demonstrated that allergen contact not only causes a clinical reaction but also leads to boosts of allergen-specific IgE responses, which can be measured in “convalescent sera” (see lines 246-247 and 261-262 of the revised manuscript).
My critique was poorly worded. I understand that this was not an allergen challenge study. I should have used the phrase allergen exposure. This new interpretation of rises in IgE as indicative of the allergen associated with the exposure suffers from the same problem as the interpretation of the viral serology- the time interval between the event and the convalescent serum is so long that intervening exposures become more likely.
The evaluation of rhinovirus infection is also problematic. Serologic assays may have a role in the assessment of rhinovirus infection but specific diagnosis with PCR assays at the time of infection is readily available and would be much preferred. The use of a panel of VP1 antigens with an interval of as long as 30 weeks (median 11 weeks) before collection of the convalescent serum raises the possibility that that the detected infection could be unrelated to the wheezing episode. The finding that 87% of patients had a response on the RV panel with only 30 serotypes tested is higher than expected and also suggests that infections unrelated to the wheezing episode may have been included. The advantage of using serology is not obvious; if the authors have a convincing argument that should be included in the manuscript.
Response: We agree with the reviewer that PCR tests can demonstrate the presence of virus-derived nucleic acid but these tests do not necessarily prove that the detected virus had caused an infection and was responsible for clinical symptoms (see: Pavia, A. T. Viral infections of the lower respiratory tract: old viruses, new viruses, and the role of diagnosis. Clin. Infect. Dis. 52, 284–289 (2011)). It should be also mentioned that RV RNA was detected in large numbers of asymptomatic infants and children (Advani, S., Sengupta, A., Forman, M., Valsamakis, A. & Milstone, A. M. Detecting respiratory viruses in asymptomatic children. Pediatr. Infect. Dis. J. 31, 1221–1226 (2012); van der Zalm, M. M. et al. Respiratory pathogens in children with and without respiratory symptoms. J. Pediatr. 154, 396–400 (2009); Winther, B., Hayden, F. G. & Hendley, J. O. Picornavirus infections in children diagnosed by RT-PCR during longitudinal surveillance with weekly sampling: association with symptomatic illness and effect of season. J. Med. Virol. 78, 644–650 (2006)). There are at least two advantages of measuring increases of RV-specific antibodies over PCR testing. First, serology demonstrates that an RV infection has indeed occurred and affected the immune response. Second, the use of micro-arrayed peptides from RV strains allows species-specific diagnosis. We have mentioned these considerations in our revised manuscript but also mentioned as a possible limitation of our study, that results obtained for samples collected longer than 10 weeks (i.e., the optimal time for measuring increases of RV-specific antibody responses, see: Niespodziana K, Cabauatan CR, Jackson DJ, et al. Rhinovirus-induced VP1-specific antibodies are group-specific and associated with severity of respiratory symptoms. EBioMedicine 2014;2(1):64-70) after the wheezing attack maybe due to a later infection not related to the wheezing episode (see lines 338-340 and 349-360) of the revised manuscript).
It is true that detection of a virus in the respiratory tract is not de facto evidence that the viral infection is the cause of the clinical syndrome in the patient. However, absent contamination or laboratory error, the detection of viral nucleic acid in the respiratory sample is definitive evidence of infection. The fact that some individuals have asymptomatic infections is irrelevant.
I agree that a serotype-specific antibody response is clear demonstration of a RV infection with that serotype. Unfortunately, type specific antibody responses are inconsistent so that the absence of antibody response is not evidence for absence of infection.
10 weeks is not the optimal time for measuring adaptive responses to RV infection. These responses are seen consistently by 3-4 weeks after infection.
Comparison of the serologic responses to rhinovirus and the allergens in a non-wheezing control group would perhaps be beneficial to the interpretation of this study.
Response: We thank the reviewer for this comment. Unfortunately, such a group was not available and we have mentioned this as a possible limitation of our study (see lines 358-359 of the revised manuscript). However, the fact, that increases of RV-specific antibodies were not detected in all tested children indicates the specificity of the microarray. Also we have shown in an earlier exposure study that the magnitudes of increases of RV-specific antibodies are related with the severity of respiratory symptoms (Niespodziana K, Cabauatan CR, Jackson DJ, et al. Rhinovirus-induced VP1-specific antibodies are group-specific and associated with severity of respiratory symptoms. EBioMedicine 2014;2(1):64-70). These considerations were also added to the revised manuscript (see lines 341-348).
Specific Comments:
Line 102: Little information about the research subjects is included. Were any virologic assays done for detection of infection at the time of the wheezing diagnosis?
Response: Following the reviewer’s suggestion we included now the information on virologic assays in the revised version of the manuscript. It should be noted that PCR-based evidence for RV infections was obtained for the majority of children who showed increases of RV-specific IgG levels by serology (see lines 117-124).
Line 151: This table is mostly blank.
Response: We are very sorry that the reviewer received a blank table. We have now included the complete Table I as a picture in the revised manuscript (see lines 174-175).
Line 162: The data provided in this paragraph seem mostly meaningless. Diagnostic serologic responses to rhinovirus infection depend on serotype-specific antibody increases. It is not clear what can be learned by a description of aggregate antibody rises to rhinovirus-species.
Response: We must respectfully disagree with the reviewer. It has been shown that cumulative RV-specific antibody responses are useful parameters (see: Megremis S, Niespodziana K, Cabauatan C, et al. Rhinovirus species-specific antibodies differentially reflect clinical outcomes in health and asthma. Am J Respir Crit Care Med 2018;198(12):1490-1499). In fact, the microarray used in this study contained 30 N-terminal VP1 peptides including 18 peptides of RV-A species, 9 peptides of RV-B and 3 peptides of RV-C species. Due to the fact, that peptides from RV strains belonging to the same RV species show higher sequence similarity than those from unrelated species, cross-reactivity to peptides from one species can be observed to establish cumulative species-specific antibody levels. This was explained in the revised manuscript (see lines 164-168).
The authors have misused their own reference. That antibody responses may correlate with number of infections over the previous year is perhaps not surprising. This has nothing to do with the use the antibody assay to associate a RV infection with a specific illness.
Line 199: Figure 2 is quite large and difficult to follow (especially since it appears to be written in Greek). Would perhaps be suitable for inclusion in the supplemental material if translated.
Response: We feel sorry that the reviewer received Figure 2 in a low quality. Unfortunately, we were not aware about that the submitted manuscript was converted in an unusual format. We have replaced the figure with a new Figure 2 and suggest to leave it in the main manuscript because it reveals the details of the findings (see lines 221-222).
Line 214: It is not clear how evidence for allergen exposure after the acute wheezing episode can be used as an indication that sensitization is a trigger for the wheezing.
Response: We thank the reviewer for this comment and refer to our above response “A retrospective identification of a particular allergen molecule as possible trigger factor for a clinical reaction can be assessed by the boost of the allergen-specific IgE levels in a follow up serum sample obtained some weeks later as shown in Figure S1 of our manuscript. We have revised our manuscript to explain the design of the serological analysis and quoted earlier studies which have demonstrated that allergen contact not only causes a clinical reaction but also leads to boosts of allergen-specific IgE responses which can be measured in “convalescent sera” (see lines 246-2247 and 261-262 of the revised manuscript).
Reviewer 3 Report
All changes have sufficiently be done. I think the paper is ready for publication.
Reviewer 4 Report
The authors have been very responsive to my earlier suggestions. This includes correcting Table 1 and improving the legibility of Figure 2. In addition, they have incorporated Supplemental Figure 1 into the manuscript as a new figure (Figure 4). Editing of the final paragraph of the results section has improved readability significantly.
Regarding patient #44, whose IgE levels rise upon follow up, the amount of increase for this patient was much smaller than for the other 3 patients and raises questions about as to if this is simply due to noise in the assay. Similarly, the majority of children who became positive for IgE to one or more allergens exhibited rather low levels of IgE. Is it possible that some of these results may be false positives? Is there a way to exclude this? For example, if the results were repeated or performed in duplicate?